# Psiadin and plectranthone selectively inhibit colorectal carcinoma cells proliferation *via* modulating cyclins signaling and apoptotic pathways

Khaled Y. Orabi[1☯‡*], Mohamed S. Abaza[2☯], Yunus A. Luqmani[1], Rajaa Al-Attiyah[3]

1 Department of Pharmaceutical Chemistry, Faculty of Pharmacy, Kuwait University, Safat, Kuwait,
2 Department of Biological Sciences, Faculty of Science, Kuwait University, Safat, Kuwait, 3 Department of Microbiology and Immunology, Faculty of Medicine, Kuwait University, Safat, Kuwait

☯ These authors contributed equally to this work.
‡ KYO is the principal author.
* kyorabi@hsc.edu.kw

**Data Availability Statement:** All relevant data are within the manuscript and S1 and S2 Raw images, S1 Table.

## Abstract

Three scarce terpenes, psiadin, plectranthone and saudinolide, were obtained after chromatographic isolation and purification from the aerial parts of the respective plants. Their identities were established based on their spectral data. Their anticancer effects against two human colorectal carcinoma cell lines, CCL233 and CCL235, along with the potential molecular mechanisms of action, were explored. Psiadin and plectranthone exhibited marked growth inhibition on both cell lines in a time- and dose-dependent manner with minimal cytotoxicity against normal breast cells (HB2). The terpenes even showed superior activities to the tested standards. Flow cytometry showed apoptosis induction and alteration in the cell cycle in colorectal cancer cells treated with both compounds. Nevertheless, it was also found that both compounds inhibited NF-κB transcriptional activity, induced mitochondrial membrane potential depolarization and increased the percentage of reactive oxygen species in the treated cancer cells in a dose-dependent manner as well. Since the anticancer effect of psiadin on cancer cells was higher than that produced by plectranthone, only psiadin was tested to determine its possible targets. The results suggested a high degree of specificity of action affecting particular cellular processes in both cancer cells. In conclusion, both terpenes, in particular psiadin, showed significant discriminative therapeutic potential between cancer and normal cells, a value that is missing in current chemotherapies.

## Introduction

Cancer is a huge health burden affecting almost every region worldwide, with estimations of 28 million new cases and a predicted 16 million cancer deaths by 2040 [1]. Colorectal cancer is the second most common cancer in women and the third in men, and accounts for about 9% of all cancer deaths [1]. During the past decades, the development of chemotherapeutics has

**Funding:** KY Orabi: this project was financially supported by Kuwait University Research sector (http://www.ovpr.ku.edu.kw/index.php/en/publicationsen/researchsectoren) through research grant number PC01/12. The spectral analyses were obtained at the Research Sector Project Unit (RSPU), Faculty of Science, Kuwait University, supported by grant numbers GS01/01 and GS01/03. Protein expression analysis was done at the Research Unit for Genomics, Proteomics and Cellomics Studies (grant number SRUL02/13). The funders had no role in study design, data collection and analysis, decision to publish, or preparation of the manuscript.

**Competing interests:** The authors have declared that no competing interests exist.

been considerably hampered by the limited sources of chemical scaffolds. However, natural products, being a rich source of chemo- and bio-diverse nutraceuticals, have been extensively explored and have led to remarkable successes. This is particularly evident in the field of cancer therapy, where almost 70% of currently used anticancer drugs are either natural products or natural product-derived compounds, such as vincristine, camptothecin and paclitaxel [2, 3]. Between 1981 and 2019, about 250 such chemical entities were approved as anticancer drugs [4]. As the largest class of natural products, terpenoids consist of approximately 25,000 compounds with potential practical applications in fragrance, flavors, chemical and pharmaceutical industries [5]. Terpenes, including sesquiterpenes and diterpenes, were previously reported to have antiproliferative effects against various types of cancer including colorectal cancer [6].

Several genera of the Families Euphorbiaceae, Lamiaceae, and Asteraceae, among many others, are endogenous to the Arab peninsula and have been excessively used in the folkloric medicine for relieving various ailments [7–10]. In this study, we focused on three plants to isolate three scarce terpenes; psiadin, plectranthone and saudinolide (Fig 1A).

Many compounds, including the kaurane diterpene psiadin, were isolated from aerial parts of *Psiadia arabica* Jaub. et Spach. (Asteraceae) [11], with only flavonoids reported to have bioactivity [12]. Other reports, however, documented different activities, like antimicrobial [13, 14], antiviral [15, 16], antiplasmodial [17, 18], and antiinflammatory activities [18, 19], for crude extracts and essential oils of different *Psiadia* species.

*Plectranthus* species was shown to contain sesquiterpenes, diterpenes and triterpenes [20]. *Plectranthus cylindraceus* is documented as a folkloric medicine in different regions of Arabia. Plectranthone, a sesquiterpene, was identified in *Plectranthus cylindraceus* Hochst. ex Benth (Lamiaceae) [20].

Different diterpenes, including saudin and saudinolide, were isolated from *Clutyia richardiana* L. (Euphorbiaceae). Saudin was reported to have a significant hypoglycemic effect in non-alloxanized fasting mice [7, 8].

We report the chromatographic isolation and spectral identification of these three terpenes and the determination of their anticancer effects on two human colorectal carcinoma cell lines. It is shown that psiadin is the most potent of the three terpenes followed by plectranthone. At equivalent concentrations, they also induced greater rates of cell death than several drugs currently utilized in cancer chemotherapy, through various mechanisms that are discussed.

## Materials and methods

### Plant material

The three plants; *Psiadia arabica* Jaub. et Spach. (known locally as "Tabbak"), *Plectranthus cylindraceus* Hochst. ex Benth (known locally as "Al-Shar") and *Cluytia richardiana* L. (known locally as "Sa'eer") were collected between March and April 2012 from Wadi Al Uss, near Abha, Saudi Arabia. The plant identities were authenticated by Dr. Sultan-ul-Abdeen and voucher specimens (# 10331, 10352 and 10362, respectively) were deposited at the herbarium of the College of Pharmacy, King Saud University, Riyadh, Saudi Arabia. All plants names have been checked with http://www.theplantlist.org.

### Extraction and isolation

**Psiadia arabica.** Two kilograms of the plant's dried aerial parts were extracted, using a Soxhlet extractor, in chloroform for 72 h. The dried greenish residue (260 g) was partitioned between chloroform and water (3 × 1 L). The chloroform layer, after evaporation, was partitioned between 10% aqueous methanol and 3 portions of *n*-hexane (3 × 1 L). The combined

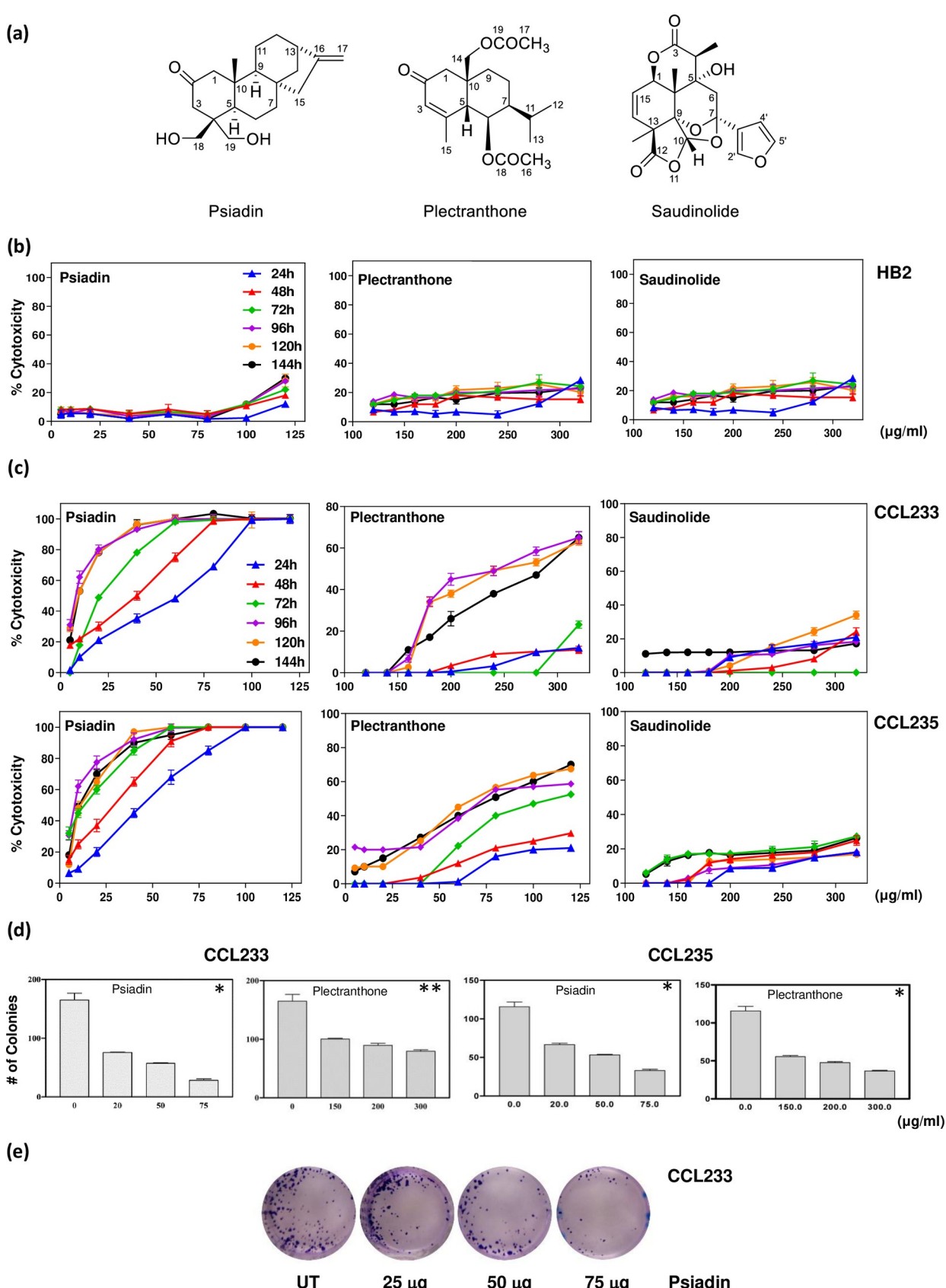

**Fig 1. Psiadin, plectranthone and saudinolide induce cytotoxicity in colorectal cancer cells.** (a) Chemical structures of the isolated terpenes. (b) Time- and dose-dependent effects of psiadin, plectranthone and saudinolide on normal human breast cells (HB2). Cells were plated into 96-well plates (27 x 103 cells/well) and incubated at 37°C in a CO2 incubator. After 18 h, cells were treated with the indicated concentrations of each terpene for the indicated periods of time (24–144 h). Cell density was measured by MTT assay. (c) Time- and dose-dependent effects of terpenes on CCL223 and CCL235 cells. Cells were plated (27 x 103 cells/well) into 96-well plates and incubated at 37°C in a CO2 incubator. After 18 h, cells were treated with the indicated concentrations of each terpene for the indicated periods of time (24–72 h). Cell density was measured by MTT assay. (d) Colony formation by untreated and terpene-treated CCL233 and CCL235 cells. Cells were plated into 24-well plate (1.0 x 105 cells/ml) in a CO2 incubator for 18 h. Cells were either left untreated or treated with the indicated concentrations of each terpene, and incubated for 24 h. Cells were then trypsinized, counted, and plated at 500 cells/ml into a 24-well plate and incubated in a non-CO2 incubator for 10–14 days. Cells were fixed in methanol and stained with 0.1% crystal violet. The stained colonies were counted and compared with an untreated control. Data are reported as the means ±SEM of three independent experiments, each one was done in triplicate. Asterisks denote significant difference from the control/vehicle, with p>0.01 (*) and p = 0.05 (**). (e) A representative image of colony formation assay using CCL233 cells and different concentrations of psiadin (magnification power 20x).

methanol layers, after evaporation, afforded 185 g as a gummy residue. This residue was chromatographed over a silica gel column (1100 g, 50 × 7.5 cm) using CHCl$_3$-MeOH, 4:1, yielding 1380 mg of the impure diterpene psiadin, which upon crystallization from methylene chloride-EtOAc gave 813 mg colorless needles of pure psiadin.

**Plectranthus cylindraceus.** One kilogram of the dried aerial parts of the plant was percolated at room temperature with 95% EtOH (3 × 4 L). The dried residue (22 g) was chromatographed over silica gel (500 gm, 4 × 130 cm), using increasing concentrations of EtOAc in *n*-hexane as an eluent, to yield, after crystallization from EtOAc-petroleum ether, 1400 mg of plectranthone as colorless prisms.

**Cluytia richardiana.** Likewise, 1.6 kg of the plant were percolated successively at room temperature in petroleum ether (60–80°C) and then in EtOAc. The EtOAc fraction, after evaporation, afforded 26 g of a greenish-white precipitate. This precipitate was subjected to chromatographic purification over flash silica gel (2.6 kg), and eluted with petroleum ether-EtOAc (8.5:1.5) to yield, after crystallization from the same solvent, 450 mg of saudinolide as colorless granules.

## Identification

Melting points were determined in open capillary tubes using a Mettler 9100 electrothermal melting point apparatus and were uncorrected. IR spectra were recorded using a Jasco FTIR-4100 spectrophotometer. UV spectra were measured in MeOH using a UV-160 IPC spectrophotometer. $^1$H and $^{13}$C NMR spectra were obtained on a Bruker Avance II 600-MHz spectrometer operating at 600 and 150 MHz, respectively. Both $^1$H and $^{13}$C NMR spectra were recorded in CDCl$_3$, and the chemical shift values were expressed in δ (ppm) relative to the internal standard TMS. Carbon multiplicities were determined using DEPT angles at 45°, 90° and 135°. 2D NMR data were obtained using the standard pulse sequence of the Bruker 600 for COSY, HSQC and HMBC. High-resolution EIMS were obtained using a double-focusing magnetic sector mass spectrometer (GC-MS DFS, Thermo).

## Cell lines

Human colorectal carcinoma cells CCL233 and CCL235, and normal human breast cells HB2 were obtained from the American Type Culture Collection (ATCC; VA, USA). The cancer lines were cultured in 90% Leibovitz's L15 medium supplemented with 10% heat-inactivated fetal bovine serum (FBS) and grown at 37°C in a non-CO$_2$ incubator. The HB2 cells were cultured in Eagle minimum essential medium (EMEM) supplemented with 10% FBS and grown at 37°C in a CO$_2$ incubator.

## Chemicals and reagents

Trypsin, Leibovitz's L-15, EMEM, FBS, and penicillin/ streptomycin solution (100×) were obtained from Mediatech (Herndon, VA, USA). Annexin V-FITC apoptosis detection kit was obtained from BD Hoffmann-La Roche Inc. (NJ, USA). DNA extraction kit was obtained from Beckman & Coulter (FL, USA). A nuclear/cytosol fractionation kit was obtained from BioVision Inc. (Mountain View, CA, USA). Antibodies used in this project and their sources are listed in S1 Table. All other reagents were obtained from Sigma Chemicals (MO, USA).

## MTT assay

Cell viability was measured at different time points (24–144 h) using MTT assay, based on the conversion of MTT to formazan crystals by mitochondrial dehydrogenases [21]. Briefly, cells were cultured at $27 \times 10^3$ cells/well in 96-well plates and incubated at 37˚C for 24 h. Then, cells were treated with increasing concentrations of the tested terpenes (or 0.2% DMSO as a vehicle control) over different time intervals (24–144 h). MTT solution (5 mg/ml) was added to the cell wells (20 μl/well); cells were then incubated for another 4 h and the medium was discarded. DMSO (200 μl) was added to each well, and the absorbance measured in a microplate reader at 492 nm. All samples were assayed in triplicate in three independent experiments. The results (mean ± SEM) are expressed as % cytotoxicity.

In separate experiments, the effects of the tested terpenes were compared with some clinically available anticancer drugs; camptothecin (CPT), 5-fluorouracil (5FU), doxorubicin (DOX) and ellipticine (ELP) over 72h of drug exposure.

As we later determined that the terpenes had effects on mitochondrial function and, therefore, might affect the MTT assay, which depends on a reaction occurring within the mitochondria, we also used the MTX assay as well as doing direct cell counting for a number of drug-treated samples and obtained similar results. Therefore, we concluded that the reaction involving MTT was not affected.

## Colony-forming assay

Briefly, cancer cells ($10^5$/well) were plated in a 24-well plate and incubated at 37˚C for 18 h. Later, cells were treated with increasing concentrations of the tested terpenes (or 0.2% DMSO as a control) and incubated at 37˚C for 24 h. Then, cells were collected, washed with Hank's Balanced Salt Solution (HBSS), counted and plated in 24-well plates at 500 cells/well, and incubated at 37˚C for 10–14 days. The resultant colonies were washed with cold phosphate-buffered saline (PBS), fixed with 100% methanol, and stained with 0.1% crystal violet. The colonies were counted using an inverted microscope.

## Cell cycle analysis

The distribution of cell cycle phases ($G_0/G_1$, S, and $G_2/M$) was determined using flow cytometry by measuring the DNA content of nuclei labeled with propidium iodide as described previously [22]. Briefly, cancer cell lines were plated at $2.5 \times 10^5$ cells/ml in 24-well plates and incubated for 18 h at 37˚C. Cells were then treated with psiadin (100–400 μg/ml), plectranthone (200–400 μg/ml) or DMSO for 24 h. Cells were collected by trypsinization, washed with cold PBS and counted. Cells were processed using a DNA-prep kit (Beckman & Coulter, USA), where cells were treated with a cell-membrane permeabilizing agent followed by propidium iodide and RNase. The samples were incubated at room temperature for 15 min before analysis by flow cytometry (FC500, Beckman & Coulter, USA). The percentages of cells in

different cell cycle phases were calculated using the statistical software package advanced DNA cell-cycle software from Phoenix Flow System (San Diego, USA).

## Apoptosis assay

Induction of apoptosis was determined by annexin V-FITC apoptosis detection kit (BD Hoffmann La Roche Inc, USA) according to the manufacturer's instructions. Briefly, cancer cells were plated ($2.5 \times 10^5$ cells/ml) in a 24-well plate and incubated at 37˚C for 18 h. Following treatment with psiadin (100–400 μg/ml), plectranthone (200–400 μg/ml) or DMSO for 24 h, cells were re-suspended in 100 μl of annexin V fluorescein and propidium iodide in HEPES buffer. Following incubation in darkness at room temperature for 15 min, cells were analyzed by flow cytometry.

## Measurement of mitochondrial membrane potential

To investigate if mitochondrial damage occurs early during the apoptosis, changes in the mitochondrial membrane potential ($\Delta\Psi$m) were measured using flow cytometry with NIR Mitochondrial Membrane Potential assay kit (ab112149, Abcam, UK). Cancer cells were plated at $2.5 \times 10^5$ cells/ml in 24-well plates for 18 h at 37˚C, followed by treatment with psiadin (50, 100 μg/ml) or plectranthone (200, 400 μg/ml) for 24 h. The adherent cells were gently detached with 0.5 mM EDTA and washed once with serum-containing media. MitoNIR Dye (5 μl) was added and the cells incubated at 37˚C for 15–30 min. Cells were, then, centrifuged at 1000 rpm for 4 min, re-suspended in 1ml of the assay buffer and the intensity of the produced fluorescence was measured in the FL4 channel (Ex / Em = 635/660) using a flow cytometer gated on the cells of interest. Experimental conditions were standardized using the mitochondrial uncoupler CCCP from Abcam (ab141229) as a positive control.

## NFκB DNA binding activity

Colorectal cancer cells were plated at $2.5 \times 10^5$ cells/ml into 24-well plates at 37˚C for 18 h, then treated with psiadin (50, 100 μg/ml), plectranthone (200, 400 μg/ml) or DMSO for 24 h. Cells were harvested and their nuclear extracts prepared using a cytosol/nuclear fractionation kit from BioVision (CA, USA) as per the manufacturer's instructions. NF-κB DNA binding activity was determined using an NF-κBp65 Transcription Factor assay kit (ab133112; Abcam, UK). Nuclear extracts were then added to the wells and incubated with shaking for 1h at room temperature. After washing with wash buffer, NF-κB specific primary antibody was added to the wells for 1h, washed, then the addition of goat anti-rabbit HRP-conjugated secondary antibody for another 1h. After further extensive washing and incubation for 15–45 min with the developer solution, the absorbance at 450 nm was determined in a plate reader.

## Measurement of reactive oxygen species generation

Cells were plated at $2.5 \times 10^5$ cells/ml into 24-well plates at 37˚C for 18 h. Media were removed and cells gently washed with HBSS (2–3 times), followed by the addition of 100 μl of 1x DCFH-DA (2',7'-dichlorofluorescein diacetate) (Sigma, USA) in PBS for 30–60 min in the dark at 37˚C. The solution was then removed and cells were washed with HBSS. The DCFH-DA-loaded cells were then treated with psiadin (50, 100 μg/ml), plectranthone (200, 400 μg/ml) or DMSO for 24 h. Cells were then examined by inverted fluorescence microscopy.

## Western blot analysis

Cells were cultured in 6 well plates ($5 \times 10^5$ cells/ml) to approximately 75–80% confluency and then treated with 75 μg/ml psiadin or DMSO. After 24h exposure, cells were harvested and

recovered by centrifugation for 5 min at 1000$x$ g. Fresh cell pellets were washed with PBS and then re-suspended into 300 μl of homogenization buffer containing 50 mM HEPES, 50 mM NaCl, 5 mM EDTA, 1% Triton X-100, 10 μg/ml leupeptin, 10 μg/ml aprotinin and 100 μg/ml PMSF. The Pierce BCA protein assay dye reagent (Pierce, Rockford, USA) was used to determine protein concentration in the cell lysate. About 30 μg of protein was mixed with an equal volume of 2$x$ sample loading buffer containing 100 mM Tris-Cl (pH 6.8), 4% (w/v) SDS, 0.2% (w/v) bromophenol blue, 20% (v/v) glycerol and 200 mM dithiothreitol, and heated at 90°C for 10 min. Lysates were loaded onto 10% SDS-polyacrylamide gels and electrophoresed at 150 V for 1 h. Proteins were then transferred onto a PVDF membrane and, using the molecular weight markers lane as a guide, cut into narrow strips (to economize on antisera) in size range of the expected protein to be detected (original gel images are presented in S1 Raw image). Each strip was then treated with 2% BSA for 1 h before being incubated overnight at 4°C with various primary antibodies (S1 Table). After removal of the antisera, the membrane was washed and subsequently incubated with anti-HRP-conjugated secondary antibody (1:500) for 1 h, and signals developed with Super Signal ECL and visualized with Kodak X-ray film. Bands were quantified by densitometry and intensity calculated proportional to that obtained for β actin on the same blot.

## Statistical analysis

Statistical analyses were performed with SPSS-25. The statistical significance of differences between the control and treated group was determined by one-way analysis of variance and Fisher's least-significant differences test. $P$ values < 0.05 were significant.

## Results and discussion

The need for new anticancer agents has prompted the search for such agents from natural resources. The potential for phytochemicals to act as anticancer agents is due to their ability to inhibit tumor growth, angiogenesis, and metastases with fewer side effects [3]. The literature abounds in examples of cytotoxic terpenes, but nothing has been mentioned about the cytotoxicity of the scarce terpenes; psiadin, plectranthone and saudinolide. This study aimed at isolating the three terpenes, investigating their cytotoxic effects on human colorectal cancer cells (CCL233, CCL235), compared with normal cells and several anticancer agents, and investigating the potential molecular mechanisms of action.

### Extraction, isolation and identification of terpenes

The three plants were collected, authenticated and voucher specimens were processed appropriately. Their coarsely powdered aerial parts were extracted and purified.

*Psiadia arabica* chloroform extract was solvent-portioned and chromatographed to yield 1380 mg of impure psiadin (Fig 1A). Crystallization from methylene chloride/ethyl acetate gave 813 mg (0.041% yield) of pure psiadin as colorless needles.

**Psiadin.** Colorless needles; mp 164–165°C; [α]$_D$ -120° ($c$ 0.1, CHCl$_3$); UV (MeOH) $\lambda_{max}$ (log $\varepsilon$) 282 (2.14) nm; IR (KBr) $v_{max}$ 3422, 1690 cm$^{-1}$; $^1$H NMR (CDCl$_3$, 600 MHz) see Table 1; $^{13}$C NMR (CDCl$_3$, 150 MHz) see Table 1; HRESMS $m/z$; 301.3257 [M-H$_2$O+H]$^+$, 283.3109 [M-2H$_2$O+H]$^+$.

IR spectrum of psiadin showed absorption bands for carbonyl and hydroxyl groups. The presence of a ketone carbonyl group was further supported from $^{13}$C NMR spectra, which exhibited 20 carbon resonances; one of them is a ketone carbonyl, which resonated as a singlet at $\delta_C$ 212.7. Its molecular formula was determined as C$_{20}$H$_{30}$O$_3$ on the basis of the ion peak at $m/z$ 319.3400 [M+1]$^+$. NMR spectral data (Table 1) showed signals characteristic of an

**Table 1.** $^1$H and $^{13}$C NMR data (CDCl$_3$) of psiadin, plectranthone and saudinolide.

| # | Psiadin | | Plectranthone | | Saudinolide | |
|---|---|---|---|---|---|---|
| | $\delta_C$, multiplicity | $\delta_H$, multiplicity ($J$ in Hz) | $\delta_C$, multiplicity | $\delta_H$, multiplicity ($J$ in Hz) | $\delta_C$, multiplicity | $\delta_H$, multiplicity ($J$ in Hz) |
| 1 | 48.6, CH$_2$ | 1$_{eq}$: 2.61, d (14.0) <br> 1$_{ax}$: 2.72, d (14.0) | 47.8, CH$_2$ | 2.06, d (15.8) <br> 2.55, d (15.8) | 73.0, CH | 5.93, m |
| 2 | 212.7, C | --- | 198.4, C | --- | --- | --- |
| 3 | 55.9, CH$_2$ | 3$_{eq}$: 2.38, dd (14.0, 1.5) <br> 3$_{ax}$: 2.06, dd (14.0, 1.5) | 128.7, CH | 5.90, br s | 172.7, C | --- |
| 4 | 43.7, C | --- | 158.0, C | --- | 47.7, CH | 2.9, dq (1.3, 8.0) |
| 5 | 50.5, CH | 1.22, m | 44.0, CH | 3.01, br s | 72.5, C | --- |
| 6 | 20.9, CH$_2$ | 6$_{eq}$: 1.65, m <br> 6$_{ax}$: 1.42, m | 71.6, CH | 5.56, br s | 42.3, CH$_2$ | 2.61, d (15.0) <br> 2.54, d (15.0) |
| 7 | 40.4, CH$_2$ | 1.56, m | 43.2, CH | 0.90, m | 108.4, C | --- |
| 8 | 44.0, C | --- | 20.9, CH$_2$ | 1.38, m <br> 1.60, m | --- | --- |
| 9 | 55.0, CH | 1.10, m | 28.8, CH$_2$ | 1.48, m <br> 1.48, m | 89.7, C | --- |
| 10 | 46.6, C | --- | 38.6, C | --- | 100.6, CH | 5.93, s |
| 11 | 18.5, CH$_2$ | 1.54, m | 29.1, CH | 1.48, m | --- | --- |
| 12 | 32.7, CH$_2$ | 1.46, m | 21.1, CH$_3$[a] | 0.88, d (6.7)[a] | 174.7, C | --- |
| 13 | 43.6, CH | 2.60, br s | 21.2, CH$_3$[a] | 0.86, d (6.7)[a] | 44.3, C | --- |
| 14 | 39.1, CH$_2$ | 14$_{eq}$: 1.84, dd (11, 1.5) <br> 14$_{ax}$: 1.10, m | 69.7, CH$_2$ | 3.86, d (11.0) <br> 4.40, d (11.0) | 126.2, CH | 5.91, d (3.9) |
| 15 | 44.7, CH$_2$ | 2.01, m | 23.2, CH$_3$ | 1.99, s | 132.3, CH | 6.22 dd (3.9, 8.0) |
| 16 | 154.7, C | --- | 21.5, CH$_3$ | 2.03, s[b] | 48.8, C | --- |
| 17 | 103.6, CH$_2$ | 4.82, s <br> 4.76, s | 21.5, CH$_3$ | 2.04, s[b] | --- | --- |
| 18 | 70.9, CH$_2$ | 3.85, s | 170.8, C[b] | --- | --- | --- |
| 19 | 65.4, CH$_2$ | 3.74, d (10.5) <br> 3.54, d (11.0) | 171.2, C[b] | --- | --- | --- |
| 20 | 19.3, CH$_3$ | 1.06, s | --- | --- | --- | --- |
| 2' | --- | --- | --- | --- | 141.9, CH | 7.68 d (1.0) |
| 3' | --- | --- | --- | --- | 124.3, C | --- |
| 4' | --- | --- | --- | --- | 108.7, CH | 6.52, dd (1.0, 1.8) |
| 5' | --- | --- | --- | --- | 144.4, CH | 7.48, t (1.8) |
| OH | --- | 5.5, br s | --- | --- | --- | 4.10, d (1.3) |
| OH | --- | 5.6, br s | --- | --- | --- | --- |
| C4-CH$_3$ | --- | ---- | --- | --- | 15.5, CH$_3$ | 1.41, d (8.0) |
| C13-CH$_3$ | --- | --- | --- | --- | 19.6, CH$_3$ | 1.19, s |
| C16-CH$_3$ | --- | --- | --- | ---- | 16.5, CH$_3$ | 1.46, s |

$a,b$ Assignments may be interchanged within the same column.

exocyclic methylene group ($\delta_H$ 4.82, 4.76, each resonated as a singlet and integrated for 1H, H-17; $\delta_C$ 103.6, t) and a single methyl group ($\delta_H$ 1.06, s, 3H, H-20; $\delta_C$ 19.3, q). Additionally, the presence of two hydroxymethyl groups was confirmed. One group (H-18) resonated at $\delta_H$ 3.85 as a singlet and integrated for 2H ($\delta_C$ 70.9, t), and the other one (H-19) resonated at $\delta_H$ 3.74, 1H, d, $J$ = 10.5 Hz, and at $\delta_H$ 3.54, 1H, d, $J$ = 10.5 Hz ($\delta_C$ 65.4, t). All the data confirmed the identity as psiadin [23]. Similarly, 95% ethanol extract of *Plectranthus cylindraceus* was

purified to afford, after crystallization from ethyl acetate/petroleum ether, 1400 mg of plectranthone (0.13% yield) as pure colorless prisms.

**Plectranthone.** Colorless prisms; mp 134–135°C; $[\alpha]_D$ -36.9° (*c* 0.03, CHCl$_3$); UV (MeOH) $\lambda_{max}$ (log $\varepsilon$) 237 (3.13) nm; IR (KBr) $\nu_{max}$ 2900, 1725, 1670 cm$^{-1}$; $^1$H NMR (CDCl$_3$, 600 MHz) see Table 1; $^{13}$C NMR (CDCl$_3$, 150 MHz) see Table 1; HRESMS *m/z*; 277.2827 [M-CH$_3$COOH+H]$^+$, 217.2472 [M-2CH$_3$COOH+H]$^+$.

The molecular formula of plectranthone was shown to be C$_{19}$H$_{28}$O$_5$ based on the presence of an ion peak at *m/z* 277.2827 [M—CH$_3$COOH+1]$^+$. IR spectrum revealed strong absorption for two different carbonyl groups, an $\alpha,\beta$-unsaturated ketone (1670 cm$^{-1}$) and an ester band (1725 cm$^{-1}$). $^{13}$C NMR spectrum (Table 1) showed 19 carbon resonances, distributed as five quartets, four triplets, five doublets, and five singlets. An isopropyl group was established based on the fact that two of the methyl carbons, resonated at $\delta_C$ 21.1 and 21.2 and, correlated with two doublets; resonated at $\delta_H$ 0.88 (*J* = 6.7 Hz, 3Hs) and 0.86 (*J* = 6.7 Hz, 3Hs). The two olefinic carbons of the $\alpha,\beta$-unsaturated ketone resonated as a singlet ($\delta_C$ 158.0, C-4) and a doublet ($\delta_C$ 128.7, C-3), with the carbonyl carbon absorbing at $\delta_C$ 198.4. $^1$H NMR spectrum of this compound exhibited two AB coupling systems. One of these was due to the protons of the isolated methylene group at C-1 and the other one was ascribed to the pair of protons at C-14. Other 2D NMR data confirmed the identity of this compound to be the eudesmane sesquiterpene plectranthone [10].

The ethyl acetate extract of *Cluytia richardiana* aerial parts, on the other hand, gave, after chromatography and crystallization, 450 mg (0.028% yield) colorless granules. which were identified as the diterpene saudinolide.

**Saudinolide.** Colorless granules; mp 231–235°C; $[\alpha]_D$ + 98° (*c* 0.142, CHCl$_3$); UV (MeOH) $\lambda_{max}$ (log $\varepsilon$) 220 (4.30), 265 (3.90) nm; IR (KBr) $\nu_{max}$ 3520 (OH), 1780 ($\gamma$-lactone), 1720 ($\delta$-lactone), 1510, 1450, 1380 (br) cm$^{-1}$; $^1$H NMR (CDCl$_3$, 600 MHz) see Table 1; $^{13}$C NMR (CDCl$_3$, 150 MHz) see Table 1; HRESMS *m/z* [MH]$^+$ 389.2544 [C$_{20}$H$_{20}$O$_8$ + H]$^+$.

Saudinolide was shown to possess the molecular formula C$_{20}$H$_{20}$O$_8$, and have a $\gamma$-lactone ($\lambda_{max}$ 1780 cm$^{-1}$; $\delta_C$ 174.7), a $\delta$-lactone ($\lambda_{max}$ 1720 cm$^{-1}$; $\delta_C$ 172.9), and a mono substituted furan ring. Additionally, $^1$H and $^{13}$C NMR data (Table 1) indicated the presence of a hydroxyl group at C-5 ($\lambda_{max}$ 3520 cm$^{-1}$; $\delta_C$ 72.5, C), a disubstituted double bond at C-14(15) ($\delta_{C-14}$ 126.2, $\delta_{C-15}$ 132.3), and an oxide bridge at C-7(10) ($\delta_{C-7}$ 108.4; $\delta_{C-10}$ 100.6). The assignment of the tertiary hydroxyl group at C-5 was aided by the presence of an AB system which resonated as a dd at $\delta_H$ 2.61 and 2.54, *J* = 15 Hz. Furthermore, a COSY experiment suggested the presence of the system -CH = CHCH(O)-, and this was confirmed by the HSQC experiment, which showed signals at ($\delta_C$ 73.0 (CH2), 44.3 (C), 126.2 (CH), 132.3 (CH), and 48.8 (C) assigned to C-1, C-13, C-14, C-15, and C-16, respectively. C-10 showed a high degree of oxygenation, as indicated by the characteristic resonances at $\delta_H$ 5.93 (s) and $\delta_C$ 100.6 (CH). This last piece of information, together with the fact that the C-7 signal occurred at $\delta_C$ 108.4, suggested the gross structure of saudinolide. Other NMR data were in accordance with those previously assigned to the diterpene saudinolide [8].

## Cytotoxic effects of the isolated terpenes on human colorectal cancer and breast normal cells

To investigate the cytotoxicity, cell viability was evaluated using MTT and colony-forming assays (Fig 1B–1E). Treatment of breast normal cells (HB2) and human colorectal cancer cells (CCL233, CCL235) with various concentrations of the tested terpenes for 24–144 h showed that psiadin, tested at concentrations of 5–120 µg/ml, had a very little effect on normal cells until the concentration exceeded 100 µg/ml when there was about 10% cytotoxicity after 24 h,

increasing to 30% with longer exposure, while plectranthone and saudinolide showed slightly greater toxicity but did not exceed 30% at the highest concentration and longest culture time assessed (Fig 1B).

On the other hand, psiadin exhibited a time- and dose-dependent cytotoxicity on both colorectal cell lines (Fig 1C). After 24 h exposure, about 50% of CCL233 cells died at 60 μg/ml (% mean toxicity = 50.4 ± 13; $P \leq 0.008$; $IC_{50}$ = 63.5 μg/ml), and after 72 h exposure, there was over 50% cell death with 23 μg/ml (% mean toxicity = 68 ± 14; $P \leq 0.002$; $IC_{50}$ = 23.1μg/ml), with a total cell kill observed with 60–75 μg/ml. Similar results were obtained with CCL235 cell line; after 24 h exposure, 50% kill was achieved at 48 μg/ml (% mean toxicity = 54.2 ± 13.9; $P \leq 0.006$; $IC_{50}$ = 48.1 μg/ml), and almost a total kill happened after 72 h at 100 μg/ml (% mean toxicity = 78 ± 10; $P \leq 0.0001$; $IC_{50}$ = 11.5 μg/ml). The ability of cells, pretreated with psiadin for 24 h, to form colonies was hampered in a dose-dependent manner starting at 20 μg/ml for both cells (for example, mean number of CCL233 treated colonies = 73 ± 1.2 *vs*. 165 ± 12 untreated one, $P \leq 0.0001$) (Fig 1D and 1E).

Plectranthone was also cytotoxic but required higher concentrations and longer exposure time; for CCL233 cells, it achieved 50% cell kill at 250 μg/ml compared with 64 μg/ml for CCL235 cells after 96 h exposure (CCL233: % mean toxicity = 32 ± 9, $P \leq 0.232$, $IC_{50}$ = 250 μg/ml; CCL235: % mean toxicity = 37 ± 6, $P \leq 0.025$, $IC_{50}$ = 64.3 μg/ml) (Fig 1C). Similarly, it inhibited colony-forming ability significantly in CCL233 cells (Fig 1D). Saudinolide, in contrast, exhibited a limited activity with a maximum growth inhibition of 20% on both cell lines (Fig 1C). In view of its lack of effect, saudinolide was excluded from further evaluation.

Furthermore, the cytotoxic effects of psiadin and plectranthone were compared to several chemotherapeutic drugs; 5-fluorouracil (5FU), doxorubicin (DOX), camptothecin (CPT) and ellipticine (ELP). The results indicated that psiadin demonstrated much higher cytotoxicity on CCL233 than the tested drugs after 24h of exposure (psiadin $IC_{50}$ was 63.5 μg/ml, while 5FU, DOX, CPT and ELP were undetected) (Fig 2A). However, after 72 h of exposure, the order of cytotoxic activities was as follows: psiadin ($IC_{50}$; 23.1 μg/ml) > CPT ($IC_{50}$; 14.6 μg/ml) > DOX ($IC_{50}$; 20.8 μg/ml) > ELP ($IC_{50}$; 106.3 μg/ml) > 5FU ($IC_{50}$; undetected) (Fig 2A).

A broadly similar pattern was observed with CCL235 cells (Fig 2A). Plectanthrone after 24 h of exposure produced a very low effect ($IC_{50}$; undetected) on CCL233. However, after 72h it achieved a cell kill of approximately 50%. This was more than with 5FU, ELP and CPT but less than with DOX. Again, a similar pattern was observed with CCL235 cells (Fig 2A).

It was concluded from the above experiments that none of the tested terpenes had appreciable toxicity against the normal cell line. Saudinolide, at the concentrations tested, was also relatively ineffective. Psiadin and plectranthone, showed appreciable activity against both cancer cell lines, with psiadin being the most effective cytotoxic agent. Additionally, both terpenes compared very favorably in their activities, particularly psiadin, showing itself to be significantly more effective than standard chemotherapy drugs such as 5FU, ELP and CPT.

In order to determine how these terpenes may exert their inhibitory actions, we looked at several cellular processes, involved in proliferation and established as hallmarks of cancer progression.

## Psiadin and plectranthone arrest the growth of colorectal cancer cells

Cell cycle arrest and apoptosis are two important mechanisms through which most anticancer drugs exert their effects [24, 25]. Many of them have been shown to arrest cell division at certain checkpoints in the cell cycle [26]. Additionally, cell cycle-mediated apoptosis has gained increasing attention, as targeting this pathway may provide a path to overcome acquired drug resistance, decrease mutagenesis and reduce toxicity [26]. In the present study, cell cycle arrest

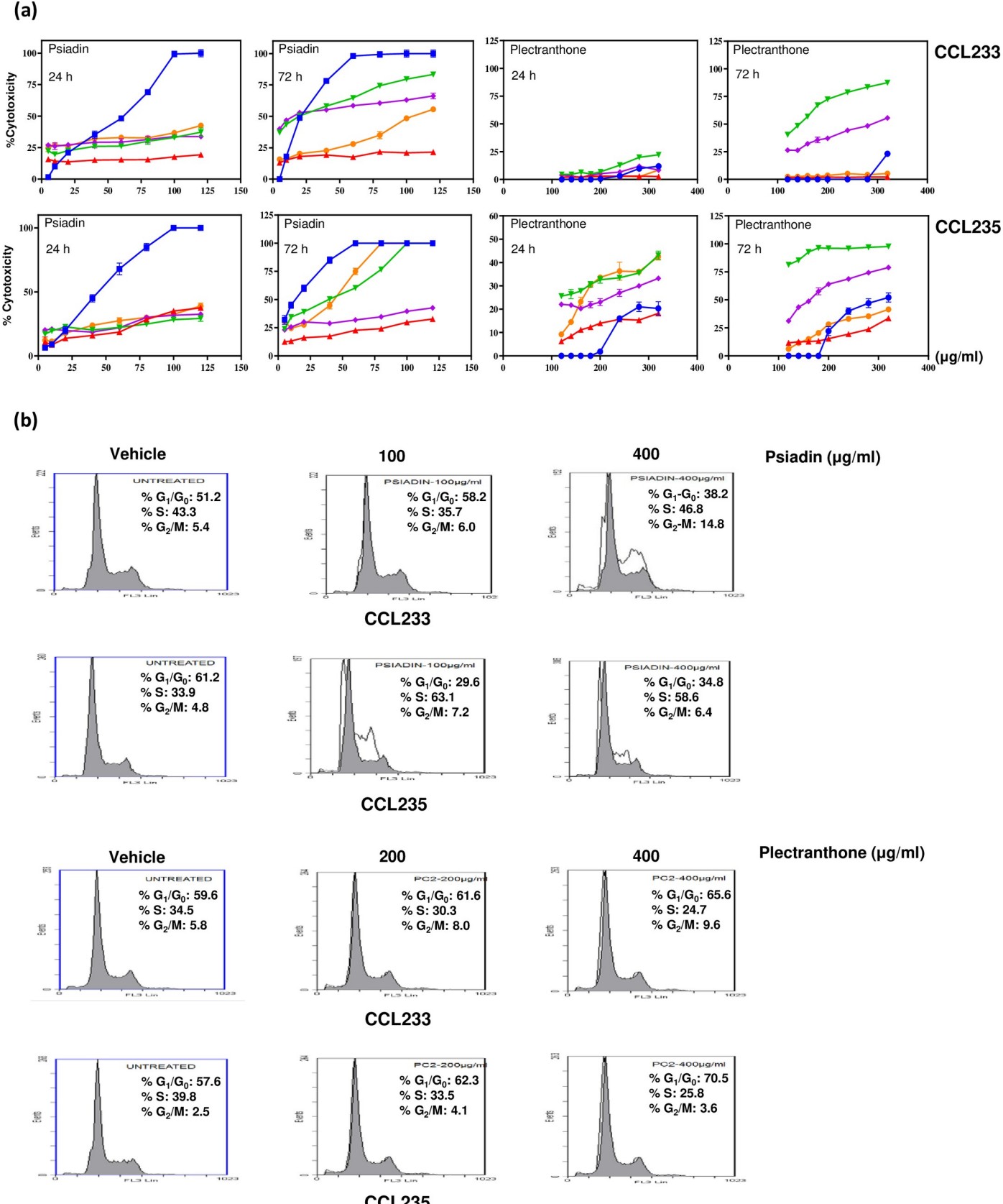

**Fig 2. (a) Psiadin and plectranthone effects on colorectal cancer cells compared to standard chemotherapeutic drugs.** Cells were plated into 96-well plates (27 x 103 cells/well) and incubated at 37˚C in a non-CO2 incubator. After 18 h, cells were treated with the indicated concentrations of psiadin (blue ■), plectranthone (blue ●), or standard chemotherapeutic drugs: 5-fluorouracil (red ▲), doxorubicin (green ▼), camptothecin (violet ♦) and ellipticine (orange ●) for 24h and 72h. Cells density was measured by MTT assay. **(b) Analysis of cell cycle stages in colorectal cancer cells treated with psiadin or plectranthone.** Cells were plated into 24-well plates (2.5 x 105 cells/ml) in a non-CO2 incubator at 37˚C for 18 h. Cells were then left untreated (control) or treated with the indicated concentrations of each terpene for 24 h. At least three samples were analyzed, and 20,000 events were scored for each sample. The vertical axis represents the relative number of events and the horizontal axis represents the fluorescence intensity.

was observed when cells were cultured with the tested terpenes. Characterization of this effect demonstrated that expression levels of cell cycle regulators were modulated by treatment with tested terpenes; arrest appeared to occur at different checkpoints in both colorectal cancer cells.

The percentage of cells in each stage of the cell cycle was determined by flow cytometry following 24 h exposure to increasing concentrations of psiadin or plectranthone (Fig 2B). Treatment of CCL233 cells with psiadin (100 µg/ml) resulted in accumulation of cells in $G_1/G_0$ phase (58.2% *vs.* 51.2% for control/untreated cells) and in $G_2/M$ phase (6.0% *vs.* 5.4% for untreated cells) with consequently fewer cells entering S phase (35.7% *vs.* 43.3% for untreated cells). With the higher dose of 400 µg/ml, there was an accumulation of cells in the S phase (46.8% *vs.* 43.3% for untreated cells), and $G_2/M$ phase (14.8% *vs.* 5.4% for untreated cells) with a corresponding decrease in the cells in the $G_0/G_1$ phase (38.2% *vs.* 51.2% for untreated cells).

With plectranthone, changes followed the same pattern, however, these were small in comparison to psiadin. A dose of 400 µg/ml resulted in the accumulation of cells in the $G_1/G_0$ phase (65.6% *vs.* 59.6% for untreated cells) and in the $G_2/M$ phase (9.6% *vs.* 5.8% for untreated cells) with fewer cells entering the S phase (24.7% *vs.* 34.5% for untreated cells).

However, treatment of CCL235 cells with various concentrations of psiadin showed a dose-dependent accumulation of the cells in both the S and $G_2/M$ phases (Fig 2B). At 100 µg/ml of psiadin the cells were arrested in both the S phase (63.1% *vs.* 33.9% for untreated cells) and $G_2/M$ phase (7.2% *vs.* 4.8% for untreated cells), with a corresponding decrease in the number of cells in $G_0/G_1$ phase (29.6% *vs.* 61.2% for untreated cells). Moreover, similar results were obtained when the used dose was increased to 400 µg/ml, where cells were accumulated in both the S phase (58.6% *vs.* 33.9% for untreated cells), and the $G_2/M$ phase (6.8% *vs.* 4.8% for untreated cells), with a corresponding decrease in the cell population in $G_0/G_1$ phase (34.8% *vs.* 61.2% for untreated cells).

Treatment of CCL235 cells with plectranthone (200 µg/ml) resulted in an accumulation of the cells in both the $G_0/G_1$ phase (62.3% *vs.* 57.6% for untreated cells) and the $G_2/M$ phase (4.1% *vs.* 2.5% for untreated cells) with a corresponding reduction in cells in the S phase (33.5% *vs.* 39.8% for untreated cells). At a higher dose (400 µg/ml), plectranthone exhibited a similar trend but with greater rates (Fig 2B).

In conclusion, plectranthone did not seem to significantly influence the cell cycle in either cell line, suggesting that its effect on the proliferation was mediated differently from psiadin. Failure to progress through cell cycle checkpoints is thought to lead to apoptosis, and as with other natural products [27, 28], this seems to be the consequence of psiadin exposure. Consistent with the MTT assay data, the viability of both cell lines was drastically reduced with concurrent increase in cells undergoing both early and late phases of apoptosis as well as direct necrosis. Again the smaller effect of plectranthone reflected its much lesser effect on cell proliferation compared with psiadin.

## Psiadin and plectranthone induce apoptosis in colorectal cancer cells

Numerous studies have shown that apoptosis is an ideal way to eradicate precancerous and/or cancerous cells [29]. However, most cancer cells block apoptosis, a mechanism through which

malignant cells survive despite the genetic and morphologic transformation. Therefore, searching for agents that can trigger apoptosis in tumor cells has become an attractive strategy for anticancer drug discovery. Here, the tested terpenes, psiadin in particular, markedly induced apoptosis in both colorectal cancer lines.

Emerging evidence from the literature has demonstrated that the anti-proliferative effects of many natural products are associated with apoptosis induction [27, 28, 30]. The results from this study are in line with these findings. Added together, dysregulation of the cell cycle mechanism and the induction of cancer cell apoptosis, are recognized as an important goal of cancer therapy.

The type of psiadin and plectranthone-induced cell death in human colorectal cancer cell lines (CCL233 and CCL235) was assessed. Cells were stained with annexin V-FITC/PI and analyzed by flow cytometry. Annexin V is a $Ca^{2+}$-dependent phospholipid-binding protein with a high affinity for PS, a membrane-bound component located on the inner surface of the cell membrane. The detection of exposed PS residues that have translocated to the cell surface is an indication of early-stage apoptosis. Annexin V assay permits the simultaneous detection of early apoptotic events, based on annexin V binding to the exposed PS, and late apoptotic/ dead events through the uptake of propidium iodide.

Treatment of CCL233 cells with psiadin (100 μg/ml) induced apoptosis, including early apoptosis (5.7% *vs*. 4.4% for vehicle), late apoptosis (61.2% *vs*. 7.1% for vehicle) and necrosis (18.3% *vs*. 1.4% for vehicle) (Fig 3A). When 400 μg/ml of psiadin were administered, only the late apoptotic stage jumped to 80.8%. For plectranthone (600 μg/ml), there was only a 22% decrease in living cells, about 6% more cells in early, and 15% in late apoptotic stages, respectively.

For CCL235 cells, psiadin reduced living cells from 87 to just 12%, increased late apoptotic cells from 6.4 to 16.3% and necrosis from 2.1% to 71.2%. Plectranthone, however, reduced living cells from 93 to about 70%, increased early apoptotic cells from 3.5 to 10.8%, late apoptotic cells from 2.7 to 14.1, and necrotic cells from 0.4% to 5.4% (Fig 3A).

## Psiadin and plectranthone induce alteration in mitochondrial membrane potential (MMP)

The change in MMP ($\Delta\psi_m$) is an important parameter of mitochondrial function and used as an indication for early apoptotic events [31]. To determine whether mitochondrial damage occurs as an early event in psiadin and plectranthone-induced apoptosis, changes in $\Delta\psi_m$ were measured using a fluorescent dye from Abcam (ab112149).

Ab112149 fluorescent dye is designed to detect cell apoptosis by measuring the loss of MMP, which coincides with the opening of the mitochondrial permeability transition pores, leading to the release of cytochrome c into the cytosol, which in turn, triggers other downstream events in the apoptotic cascade. Cells stained with the MitoNIR dye can be visualized with a flow cytometer at red excitation and far-red emission (FL4 channel). In untreated cells, the MitoNIR dye accumulates in the mitochondria. However, in apoptotic cells, the NIR stain intensity decreases due to the collapse of MMP.

Our results showed that the fluorescence intensity decreases with the increase of psiadin and plectranthone concentrations in both human colorectal cancer cell lines. This indicated that both psiadin and plectranthone could induce mitochondrial membrane potential depolarization in a dose-dependent manner (Fig 3B). The effect was more pronounced with psiadin.

It was also shown that psiadin and plectranthone exerted similar effects to the mitochondrial uncoupler (4-(trifluoromethoxy)phenyl) carbonohydrazonoyl dicyanide) in producing some reduction in the $\Delta\Psi m$ in both cell lines, indicating that these compounds induce end-

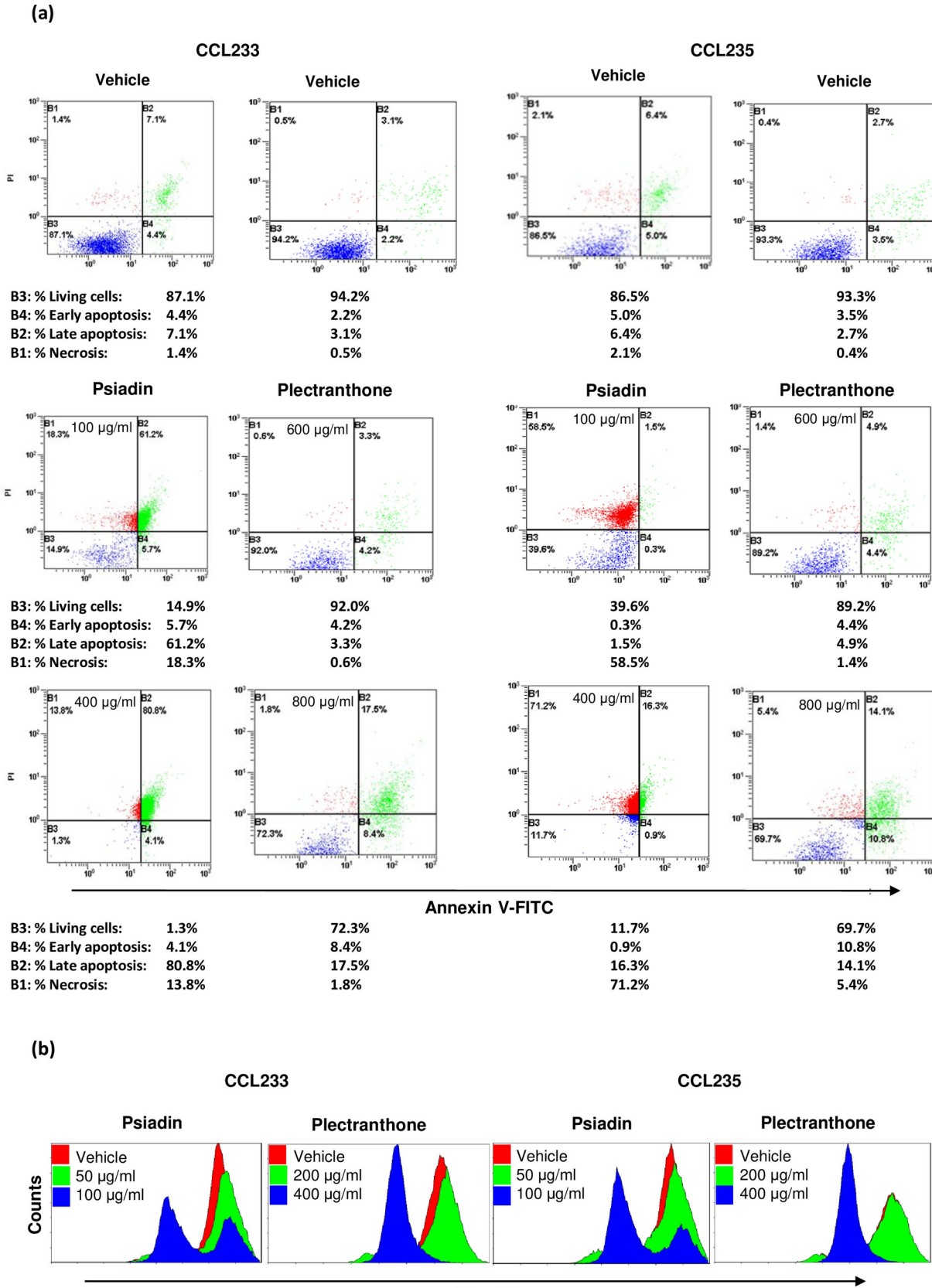

(a)

CCL233

CCL235

Vehicle

| | |
|---|---|
| B3: % Living cells: 87.1% | 94.2% |
| B4: % Early apoptosis: 4.4% | 2.2% |
| B2: % Late apoptosis: 7.1% | 3.1% |
| B1: % Necrosis: 1.4% | 0.5% |

Vehicle

| | |
|---|---|
| 86.5% | 93.3% |
| 5.0% | 3.5% |
| 6.4% | 2.7% |
| 2.1% | 0.4% |

Psiadin    Plectranthone

| | |
|---|---|
| B3: % Living cells: 14.9% | 92.0% |
| B4: % Early apoptosis: 5.7% | 4.2% |
| B2: % Late apoptosis: 61.2% | 3.3% |
| B1: % Necrosis: 18.3% | 0.6% |

Psiadin    Plectranthone

| | |
|---|---|
| 39.6% | 89.2% |
| 0.3% | 4.4% |
| 1.5% | 4.9% |
| 58.5% | 1.4% |

Annexin V-FITC

| | |
|---|---|
| B3: % Living cells: 1.3% | 72.3% |
| B4: % Early apoptosis: 4.1% | 8.4% |
| B2: % Late apoptosis: 80.8% | 17.5% |
| B1: % Necrosis: 13.8% | 1.8% |

| | |
|---|---|
| 11.7% | 69.7% |
| 0.9% | 10.8% |
| 16.3% | 14.1% |
| 71.2% | 5.4% |

(b)

CCL233    CCL235

Psiadin    Plectranthone    Psiadin    Plectranthone

Counts

FL4 - H

**Fig 3.** (a) Induction of apoptosis in colorectal cancer cells treated with psiadin or plectranthone. Cells were plated into 24-well plates (2.5 x 103 cells/ml) in a non-CO2 incubator at 37˚C for 18 h. Cells were then treated with the indicated concentrations of each terpene for 24 h. Cells were processed and stained with annexin V-FITC/PI, and then analyzed by flow cytometry as described in Methods. (b) Mitochondrial membrane potential alteration in colorectal cancer cells treated with psiadin or plectranthone. Cells were plated in 24-well plates (2.5 x 105 cell/ml) in a non-CO2 incubator for 18 h. Cells were treated with the indicated concentrations of each terpene for 24 h. Staining with MitoNIR dye was carried out as described in Methods, and the fluorescence intensity was measured in a flow cytometry in the FL4-H channel.

stage mitochondrial dysfunction. This has been observed with several common anticancer drugs as well [32]. Moreover, the monoterpene carvone was shown to be cytotoxic to colorectal cancer via modulating oxidative stress. Similar to psiadin and plectranthone, carvone was shown to collapse the mitochondrial membrane potential ($\Delta\Psi$m) [33].

## Psiadin reduces NFκB DNA binding activity of cells

Activated NF-κB participates through multiple steps in cancer resistance to chemo and radio-therapies. Animal models and *in vitro* studies have established links between NF-κB and carcinogenesis, highlighting its significance as a target in cancer treatment and chemoprevention [34]. Therefore, the potential of psiadin and plectranthone to inhibit NF-κB DNA binding activity was tested. Indeed, treatment of cancer cells with psiadin (50 and 100 μg/ml), and plectranthone (200 and 400 μg/ml) for 24 h inhibited NF-κB DNA binding activity, compared to untreated control cells in a dose-dependent manner (Fig 4A). The tested terpenes inhibited NF-κB transcriptional activity *via* inhibiting the ability of NF-κB to bind to the target genes.

The x-ray structure of RelA, a subfamily of NF-κB proteins, showed that it possesses cysteine residues in its DNA binding site, critical for optimal NF-κB protein/DNA interaction [35, 36]. Psiadin and plectranthone might have affected cysteine residues and disrupted NF-κB protein/DNA interaction. Our results are consistent with several studied terpenes. For example, avicin [37] and the kaurane diterpene, kamebakaurin [38], were shown to inhibit the binding of NF-κB to specific DNA sequences. Additionally, kamebakaurin failed to inhibit the binding of NF-κB when Cys[62], in the DNA binding site, was mutated [38], suggesting that kamebakaurin inhibited NF-κB transcriptional activity by modifying Cys[62] of NF-κB. Psiadin, another kaurane diterpene, was shown in this study to inhibit NF-κB protein/DNA binding. The inhibition of NF-κB after nuclear translocation of activated NF-κB may be a common feature of compounds with kaurane diterpene structure.

## Psiadin induces reactive oxygen species (ROS) generation

Free radicals and other reactive species play influential roles in many pathophysiological processes, including cancer. It has been reported that high levels of ROS are associated with apoptosis in cancer cells. Once produced within a cell, free radicals can damage a wide variety of cellular constituents, including DNA [39]. ROS production by human colorectal cancer cell lines treated with psiadin and plectranthone was determined by fluorescence microscopy.

As shown in Fig 4B, neither cell line under control conditions exhibited any observable fluorescence using a ROS detection kit. However, treatment with either psiadin or plectranthone resulted in the detection of intense fluorescence, in both cell populations, in a dose-dependent manner, indicating the presence of ROS.

## Protein expression

We measured, using Western blotting, the levels of various proteins that are known to be involved in cell cycle regulation and apoptotic processes, as well as some that are associated with major cell signaling pathways. For logistical reasons we limited this to psiadin as the most

**(a)**

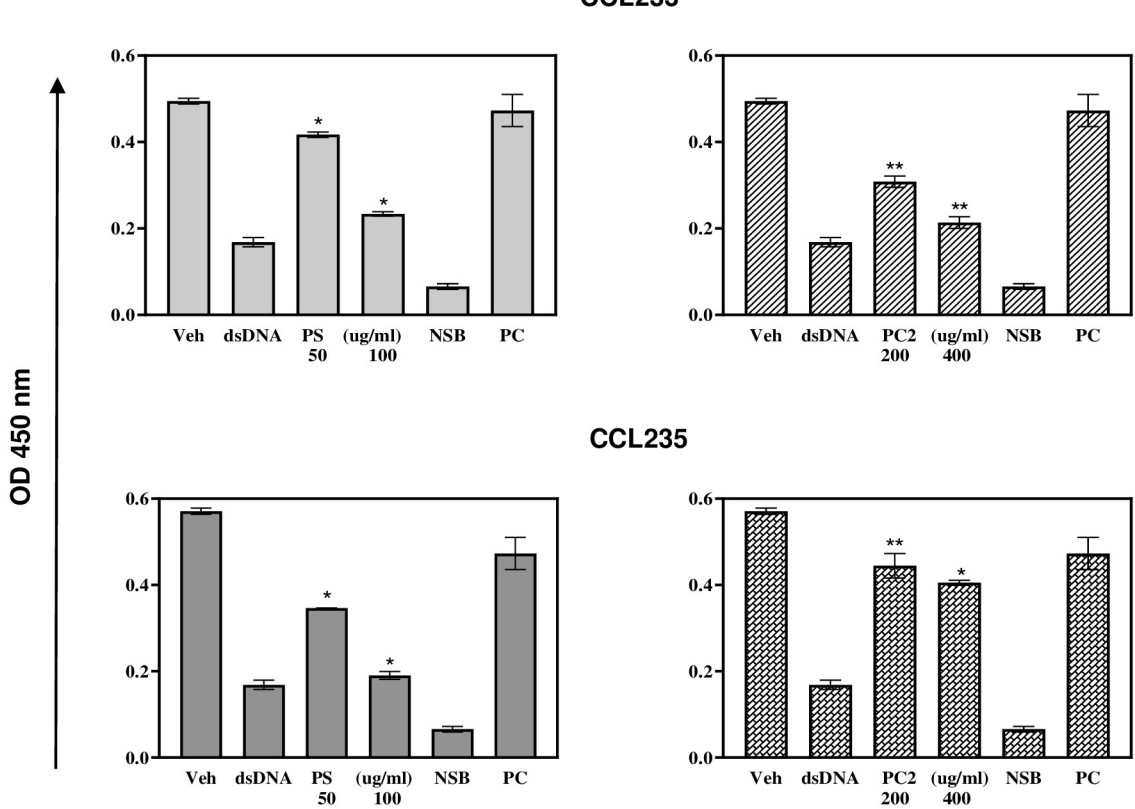

**(b)**

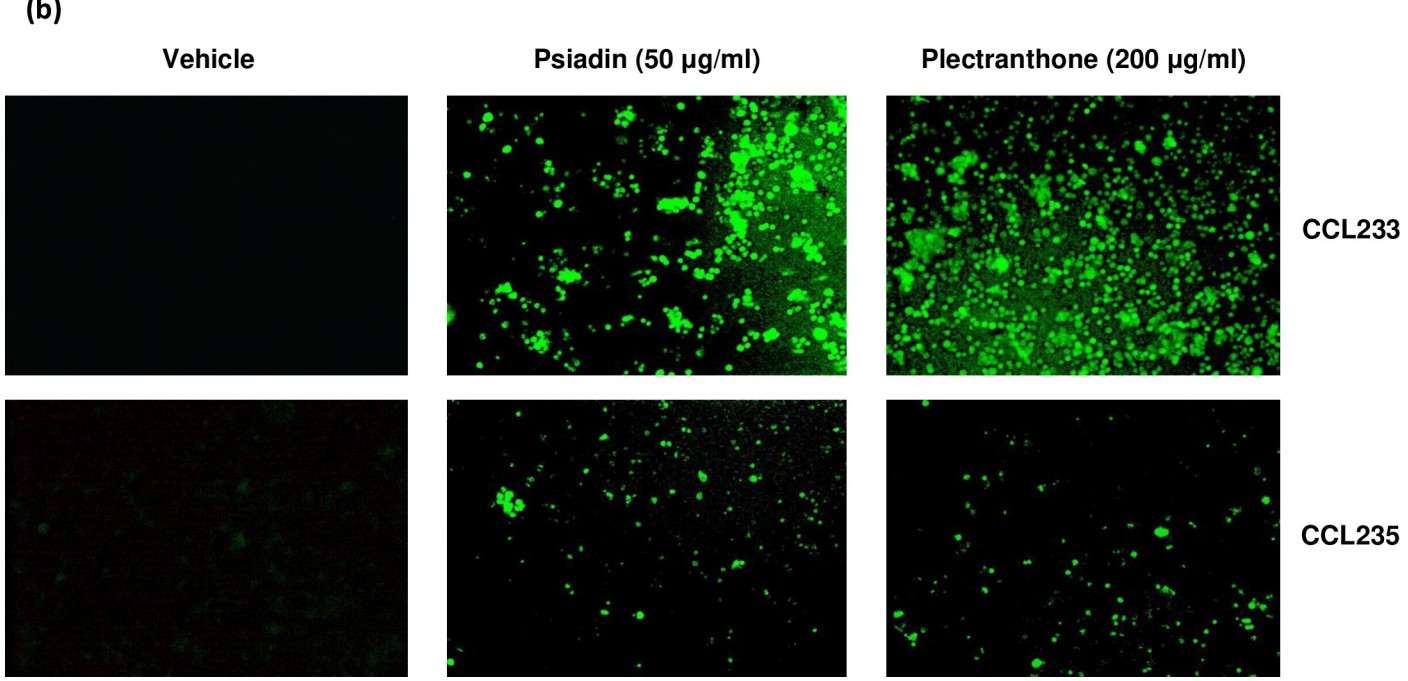

**Fig 4.** (a) NFκB DNA binding activity in colorectal cancer cells treated with psiadin (PS) or plectranthone (PC2). Cells were plated in 24-well plates (2.5 x 105 cells/ml) in a non-CO2 incubator at 37˚C for 18 h. Cells were treated with the indicated concentrations of each terpene for 24 h. Nuclear protein was extracted and tested for NFκB DNA binding activity using NFκB p65 Transcription Factor Assay Kit. NSB: non-specific binding, PC: positive control, Veh: vehicle, dsDNA: specific competitor. Drug treatments significantly differ from control/vehicle with p>0.01 (*) and p = 0.02 (**). (b) ROS generation in colorectal cancer cells treated with psiadin or plectranthone. Cells were plated into 24-well plates (2.5 x 105 cells/ml) in a non-CO2 incubator at 37˚C for 18 h. Cells were treated with the indicated concentrations of each terpene for 48 h, washed with HBSS and treated with DCFH-DA/media solution (1X, 100 μl) for 30–60 min at 37˚C. Cells were examined with inverted immunofluorescent microscope.

effective of the three terpenes under investigation in this study, and to one (CCL235) of the two cell lines since they had so far shown broadly similar behavior. This data is illustrated in Fig 5 which shows only the protein targets that were changed (data for the proteins that were analyzed but did not show any change with psiadin are not presented). The uncropped original gel images for the changed protein targets and loading control actin are provided in S1 and S2 Raw images. Fig 5 (Panel A a-f) shows the expression of proteins known to be associated with the cell cycle. Several cyclins, A2, B1, D1 and E2, as well as cyclin-dependent kinases CDK4 and CDK6 were substantially reduced in psiadin-treated cells. CDK2, on the other hand, remained unchanged. The level of p-KIP, a CDK interacting protein/kinase inhibitory protein, was also unaffected. Panel B a and b shows rises in BAK and in the level of cleaved Caspase-9. Several other pro-apoptotic factors; AIF and BID, as well as procaspase, Caspase-3 and 8 were unchanged. Among the anti-apoptotic factors (Panel C a), we observed a major reduction in MCL1, a pro-survival member of the BCL2 family. There was a small but insignificant reduction in CIAP1 and 2 and no change in FLIP.

We examined a few mitochondrial proteins and did not find much change in SMAC, and very small but insignificant increases in BCLX and Cyt c. We also looked at two well-known tumor suppressor genes (Panel D a-c). Of these, p53 levels were reduced, but there was no change in the level of the phosphorylated form. In contrast, there was a substantial reduction in the levels of both total and phosphorylated forms of the RB protein.

Fig 5 Panel E a and b shows that a key protein involved in cell signaling pathways, p44/42p-ERK, was substantially reduced; on the other hand there was an increase in the phosphorylated form of AKT (although total AKT remained unchanged), and a small but insignificant rise in c-RAF which was inconsistent, while both MEK and p-MEK didn't change. On the other hand, p38 MAPK was e reduced. Finally, Panel F, also shows a reduction in PARP which is involved in DNA repair.

Eukaryotic cells are dependent upon the production of various cyclin proteins, which through interaction with corresponding cyclin-dependent kinases, facilitate the progression of cells through cell cycle checkpoints to, ultimately, lead to cell division. Mitogenic signals such as those mediated through ERK, which was down-regulated by psiadin, result in the association of cyclin D with CDK4/6 leading to increased phosphorylation and, thereby, deactivation of the tumor suppressor RB protein [40], which then relieves its repression of transcription factor E2F allowing it to transcribe cyclins A and E which can then combine with CDK2 to take the cell through the $G_1$ and $G_2$/M checkpoints.

A general decrease was observed in the cyclins that were measured, as well as in CDK4 and CDK6 in psiadin-treated CCL235 cells, indicating a complete disruption of the cycling process. Interestingly, there was no change in either CDK2 or its interacting factor KIP, suggesting some specificity in psiadin action. Another important tumor suppressor, p53, was also reduced by psiadin but with no change in its extent of phosphorylation suggesting it was not a major target.

Apoptosis pathways can be either extrinsic and intrinsic or mitochondrial one [41], involving activation through the cleavage of the initiator Caspase-8 and -9, respectively. Activation

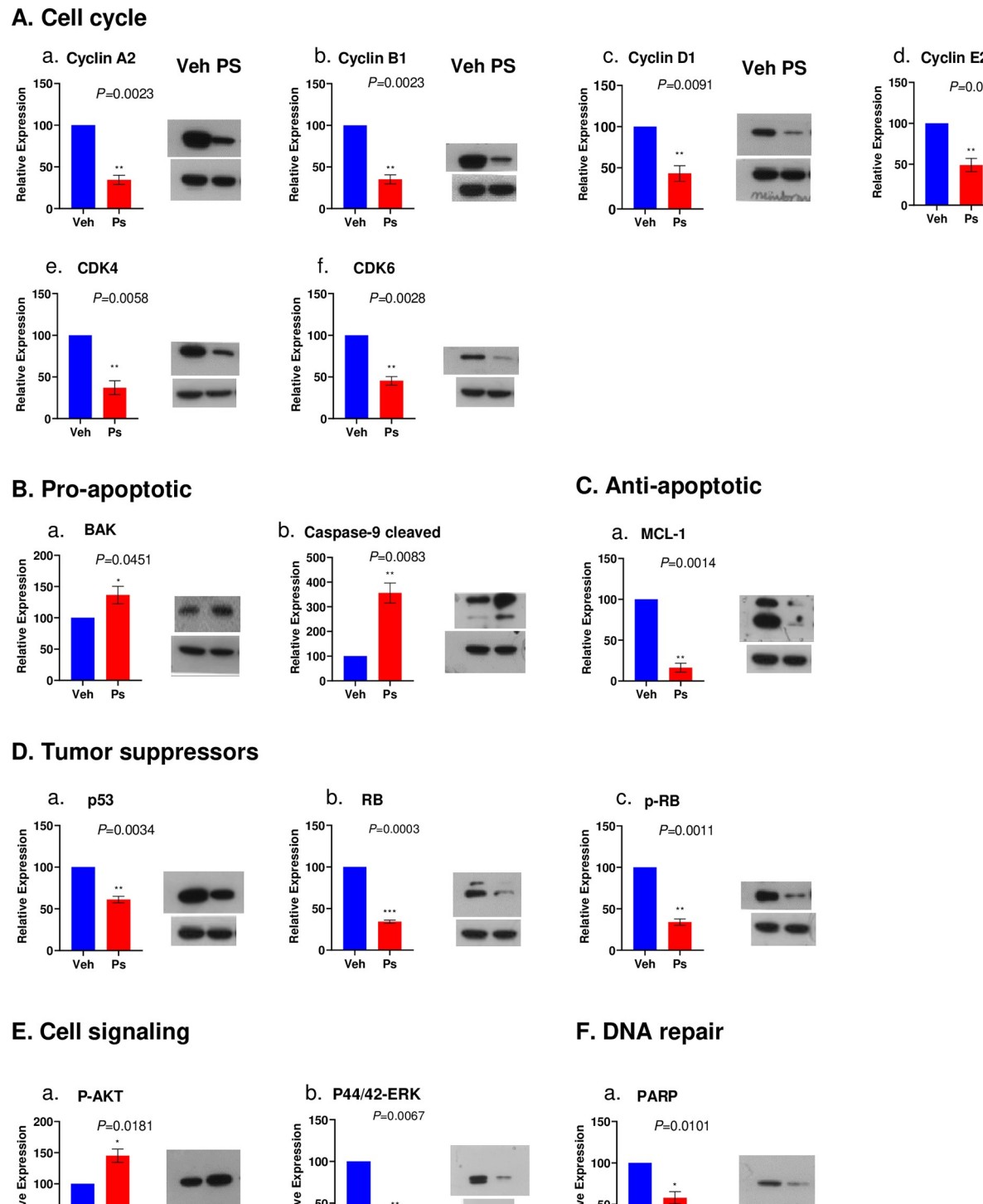

**Fig 5. Protein expression in CCL235 cells.** Proteins implicated in various cellular functions as indicated, measured in cells treated with psiadin (PS) or vehicle control (Veh). Protein extracts were electrophoresed on polyacrylamide gels, electroblotted onto PVDF membranes and incubated with appropriate primary antibodies, followed by secondary antibodies and visualized by autoradiography with ECL reagents as described in Methods. Example blot is shown for each target protein with loading control actin under it; histogram shows densitometric quantification for 3 determinations (mean ± SD) normalized to actin and with Veh set at 100.

of Caspase-9 [42] involves the formation of an apoptosome [43] (involves BCL2 family members, such as BAX and BAK) and the release of pro-apoptotic proteins, including cytochrome c [44]. Thus, psiadin in CCL235 cells elevated BAK (but not BCLX) and, substantially, increased cleaved Caspase-9 but did not affect Caspase-8, suggesting that it may preferentially affect one pathway and not the other. Activated Caspase-9 is thought to cleave Caspase-3 and -7, but we did not observe any reduction in Caspase-3 at least. At the same time, psiadin reduced the level of poly (ADP-ribose) polymerase (PARP), which is involved in the repair of single-strand DNA breaks arising from oxidative stress [45].

It has been reported that the *d*-limonene, a citrus oil-derived monoterpene, exhibits chemo-preventive and chemotherapeutic activities against various types of cancer including colorectal cancer. This was shown to be mediated via apoptosis induction. Similar to psiadin action on protein expression, limonene was reported to upregulate BAK, cleaved caspase-9 and cytochrome C. However, unlike psiadin, it was shown to upregulate PARP as well [46].

It has been reported that resistance to apoptosis may depend on the activation of NF-κB [47], *via* its many target genes [48], and its action in preventing mitochondrial- mediated apoptosis through neutralization of ROS. Consistent with this, our findings that both terpenes cause apoptosis and, significantly, reduced NF-κB binding of nuclear extracts to a consensus DNA response element for both cell lines.

Among other anti-apoptotic molecules, there was no change in FLIP but a very remarkable reduction of the MCL protein induced by psiadin. Elevated expression of MCL [49], a member of the BCL2 family, has been reported as a predictor of poor prognosis in non-small cell lung cancer [50, 51], while silencing of the MCL gene promotes senescence and apoptosis in glioma [52]. This action of MCL is thought to be mediated through inhibition of P13/AKT pathway. Although we observed no change in the total AKT, its phosphorylated form was unexpectedly increased by psiadin. However, psiadin did not affect MEK and reduced the levels of p38 and p44/42-ERK, to block proliferative pathways at those intermediates.

In conclusion, we have examined the anticancer potential of three terpenes isolated from their respective plants and shown that of these, psiadin is a very promising lead diterpene. It showed little activity against a normal cell line but was significantly more effective at killing two colorectal carcinoma cell lines than several currently used chemotherapeutic drugs. It appears to exert its actions by disrupting several cellular pathways, including cell cycle proteins, stimulation of apoptotic pathways mediated through mitochondria, very pronounced induction of ROS and inhibition in expression and function of RB and MCL proteins. Whether the latter effects are direct actions or through modulation of other intermediates remains to be seen.

## Supporting information

**S1 Table. Antibodies, their target proteins and sources.**
(PDF)

**S1 Raw image. Protein expression in CCL235 cells.** Original X-ray films of the western blot membrane strips are shown here, of the corresponding cropped images shown in Fig 5. Lanes 1 and 2 from the left are the relevant lanes that correspond to Veh and Ps-treated samples. Lanes marked with X are not included in Fig 5 and are not relevant to this study.
(PDF)

**S2 Raw image. Actin expression in CCL235 cells.** Original X-ray films of the western blot membrane strips are shown here of the actin bands (loading control) corresponding to each of the protein targets as indicated, which are shown as cropped images in Fig 5. Lanes 1 and 2

from the left are the relevant lanes that correspond to Veh and Ps-treated samples. Lanes or bands marked X are not included in Fig 5 and are not relevant to this study.
(PDF)

## Acknowledgments

The authors thank Ms. Samar Faggal for her excellent technical assistance with the protein expression analysis.

## Author Contributions

**Conceptualization:** Khaled Y. Orabi, Mohamed S. Abaza.

**Data curation:** Khaled Y. Orabi, Mohamed S. Abaza, Yunus A. Luqmani, Rajaa Al-Attiyah.

**Formal analysis:** Khaled Y. Orabi, Mohamed S. Abaza, Yunus A. Luqmani, Rajaa Al-Attiyah.

**Funding acquisition:** Khaled Y. Orabi.

**Investigation:** Khaled Y. Orabi, Mohamed S. Abaza, Yunus A. Luqmani.

**Methodology:** Khaled Y. Orabi, Mohamed S. Abaza, Yunus A. Luqmani.

**Project administration:** Khaled Y. Orabi.

**Resources:** Khaled Y. Orabi, Mohamed S. Abaza, Rajaa Al-Attiyah.

**Software:** Khaled Y. Orabi.

**Supervision:** Khaled Y. Orabi.

**Validation:** Khaled Y. Orabi.

**Visualization:** Khaled Y. Orabi.

**Writing – original draft:** Khaled Y. Orabi.

**Writing – review & editing:** Khaled Y. Orabi.

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
