## [Decision Letter · Decision Letter 0]

13 Apr 2021

PONE-D-21-06574

Psiadin and Plectranthone selectively inhibit colorectal carcinoma cells proliferation via modulating cyclins signaling and apoptotic pathways

PLOS ONE

Dear Dr. Khaled Y Orabi,

Thank you for submitting your manuscript to PLOS ONE. After careful consideration, we feel that it has merit but does not fully meet PLOS ONE’s publication criteria as it currently stands. Therefore, we invite you to submit a revised version of the manuscript that addresses the points raised during the review process.

Please submit your revised manuscript within 3 weeks time. If you will need more time than this to complete your revisions, please reply to this message or contact the journal office at plosone@plos.org. Please include the following items when submitting your revised manuscript:

We look forward to receiving your revised manuscript.

Kind regards,

Mahmood ur Rahman Ansari, Ph.D

Academic Editor

PLOS ONE

Additional Editor Comments:

1. The authors are advised to address the comments raised by the reviewers. Especially, reviewer 1 has made significant improvements in English language. Please see the attached edited copy of the article which Reviewer 1 has attached. Reviewers 2 and 3 gave valuable suggestions to improve the manuscript. Please provide pint-wise reply to each of the comment and mention changes in cover letter where needed.

2. The authors are required to submit original pictures of Gel Electrophoresis given in Figure-5 

Journal Requirements:

Reviewers' comments:

Reviewer #1: I am happy to see the elaboration of manuscript. But, it has some scientific issues in the current version, and I have highlighted issues here in the attached MS file. Therefore, the present draft should be revised accordingly.

Moreover, following reference could be cited;

https://doi.org/10.1016/j.toxicon.2020.10.012

Reviewer #2: Orabi et al., has tested the effect of various terpenes as anti-cancerous natural products. Out of three compounds tested, Psiadins were found to have promising anti cancerous effect than PC2 as measured by cytotoxicity, mitochondrial membrane potential, and ROS production. Authors have also hinted upon the involvement of NFkb downstream to the effect of selected terpenes. However, a detailed mechanism remains to be answered. Overall the study tries to address relevant translational question however looks preliminary. Some questions to address are mentioned below:

1. Why colorectal cancer cell lines are selected? Any history of specific effect of terpenes vs. colorectal cancer?

2. Are the compounds added only once and followed for observation or replenished after every 24 hrs? This is important considering varying half-life of various compounds.

3. The quality of images are not up to the mark, too difficult to follow the details. Image 1d, 3b, 4c & some blots looks similar to the authors preprint describing effect of terpenes in Hepato carcinoma cells. This also reflects that terpenes have broad anti proliferative activity against range of cell types (as reported previously in literature) .

4. Please properly rearrange the blots with labeling.

5. What is the logic of culturing cells in a non-CO2 incubator? Do authors mean hypoxic conditions to induce stress?

6. What is the IC50 of compounds? How are the doses selected to test for toxicity?

7. Why no great difference was observed on treatment of PC2 with 200 and 400 uM (Fig 1d), whereas in Fig 4a, b significant inhibition (TF activity) was observed on treatment of 400 uM of PC2.

8. Result headings should describe the major results discussed. For eg measuring ROS and protein expression are not appropriate.

9. There should be a clear demarcation of results and discussion to increase the readability of the work.

Reviewer #3: Overall, this is a concise, clear, well planned and well executed study. The manuscript is well-written. The study focused on anticancer potential of three scarce terpenes, psiadin, plectranthone and saudinolide. Their anticancer activity was tested on two human colorectal carcinoma cell lines, CCL233 and CCL235.

The presentation of data is clear and concise in text and figures. Results have been explained comprehensively and discussed critically to reach conclusions. Bibliography contains complete, accurate and up-to-date list of references with cautious care given to research ethics, including plagiarism and proper citation.This is a good quality manuscript but some MAJOR REVISIONS are required before it is accepted for publication.

PLease also include synergism (if any) between the terpenes tested in this study with standard drug.

---

## [Author Response · Author response to Decision Letter 0]

19 May 2021

Editorial Comments:

1. The authors are advised to address the comments raised by the reviewers. Especially, reviewer 1 has made significant improvements in English language. Please see the attached edited copy of the article which Reviewer 1 has attached.

Response: All suggested corrections in the attached copy of the article were done. The authors appreciate the reviewer’s genuine efforts to improve the ms quality!

2. Reviewers 2 and 3 gave valuable suggestions to improve the manuscript. Please provide pint-wise reply to each of the comment and mention changes in cover letter where needed. All points were answered point-by-point in the rebuttal letter.

3 The authors are required to submit original pictures of Gel Electrophoresis given in Figure-5, since PLOS ONE now requires that authors provide the original uncropped and unadjusted images underlying all blot or gel results reported in a submission’s figures or Supporting Information files. When you submit your revised manuscript, please ensure that your figures adhere fully to these guidelines and provide the original underlying images for all blot or gel data reported in your submission.

Response: The uncropped original gel images and the loading control actin are provided in Supporting Information S2 and S3, respectively.

Moreover, the authors made sure that all figures do adhere to the journal guidelines.

Reviewer 1

1. I am happy to see the elaboration of manuscript. But, it has some scientific issues in the current version, and I have highlighted issues here in the attached MS file. Therefore, the present draft should be revised accordingly.

Response: All suggested corrections in the MS file sent by reviewer #1 were done.

Reviewer 2

1. Orabi et al., has tested the effect of various terpenes as anti-cancerous natural products. Out of three compounds tested, psiadin was found to have promising anti cancerous effect than PC2 as measured by cytotoxicity, mitochondrial membrane potential, and ROS production. Authors have also hinted upon the involvement of NF-�B downstream to the effect of selected terpenes. However, a detailed mechanism remains to be answered. Overall the study tries to address relevant translational question however looks preliminary. Some questions to address are mentioned below.

Response: The generation of ROS plays a vital role in cellular proliferation, differentiation, and apoptosis. ROS stress is oncogenic and increases the metabolic activity [1, references used in these responses are mentioned at the end]. In the present study, ROS production after triterpene treatment was higher in psiadin-treated human colorectal cancer cells than in untreated control cells. Therefore, psiadin may be considered a potential exogeneous ROS inducer for initiating apoptosis in human colorectal cancer cells. Our results are consistent with those reported for other phytochemicals targeting different types of cancers [2, 3]. ROS eliminates cancer cells by arresting the cell cycle at various checkpoints and therefore induces apoptosis [4].

On the other hand, NF-�B is a multi-subunit transcription factor which is maintained in the cytoplasm through interaction with the inhibitors of NF-�B. Upon dissociation, NF-�B moves into the nucleus and promotes cancer cell proliferation, angiogenesis and metastasis as well as inhibits apoptosis. Many different types of cancers, including colorectal cancer, show high NF-�B activity. In the present study, the DNA-binding activity of NF-�B in colorectal cancer cells treated with psiadin was significantly reduced. NF-�B activation transcriptionally activates several pro-survival genes including c-IAP-1, c-IAP-2 and XIAP [5]. Appositive feedback loop, c-IAP2 and XIAP appear to trigger the activation of NF-�B [6]. Inhibition of NF-�B by psiadin would inhibit the pro-survival genes leading to an induction of apoptosis.

1. Why colorectal cancer cell lines are selected? Any history of specific effect of terpenes vs. colorectal cancer?

Response: Since it is the second most common cancer in women and the third in men, and accounts for about 9% of all cancer deaths, colorectal cancer cell lines were selected.

Terpenes, including sesquiterpenes and diterpenes, were previously reported to have antiproliferative effects against various kinds of cancer including colorectal cancer cells. However, none of the terpenes presented in this ms (psiadin, plectranthone, saudinolide) have been tested before on any type of cancers.

Several referenced paragraphs were added, where appropriate, to show these data.

3. Are the compounds added only once and followed for observation or replenished after every 24 hrs? This is important considering varying half-life of various compounds.

Response: The tested terpenes were added once and followed for observation as mentioned in Materials and Methods.

4. The quality of images is not up to the mark, too difficult to follow the details. Image 1d, 3b, 4c & some blots look similar to the authors preprint describing effect of terpenes in Hepato carcinoma cells. This also reflects that terpenes have broad anti proliferative activity against range of cell types (as reported previously in literature).

Response: The quality of the images in Figures 1, 2 and 5 were improved. Different colors were used with different graphs to facilitate data reading and make it easier to follow the details in these figures. Moreover, modifications were made to Fig 5 to better present the needed data.

The tested terpenes showed only broad similarity in their action on colorectal and hepatocellular carcinoma cells. 

5. Please properly rearrange the blots with labeling.

Response: The blots were properly arranged and Figure 5 was modified to better present the needed results.

6. What is the logic of culturing cells in a non-CO2 incubator? Do authors mean hypoxic conditions to induce stress?

Response: Human colorectal cancer cell lines CCL233 and CCL235 were cultivated in Leibovitz’s L15 medium (90%) and fetal bovine serum (10%). L15 medium was used with a free gas exchange with air. The standard sodium bicarbonate/CO2 buffering system was replaced by a combination of phosphate buffer, free-base amino acid, higher level of sodium pyruvate and galactose. A CO2 and air mixture was detrimental to the cells when used with this medium for cultivation. If cells in L15 were incubated with CO2, the medium could quickly turn acidic and would likely kill the culture.

7. What is the IC50 of compounds? How are the doses selected to test for toxicity?

Response: IC50 of the tested terpenes at the different exposure times are mentioned in the manuscript under “Results and Discussion” - subheading: “Cytotoxic effects of the isolated terpenes on human colorectal cancer and breast normal cells”. Dose- and time-dependent studies were used to select the doses for testing the toxicity.

8. Why no great difference was observed on treatment of PC2 with 200 and 400 uM (Fig 1d), whereas in Fig 4a, b significant inhibition (TF activity) was observed on treatment of 400 uM of PC2.

Response: Fig 1d shows the effect of PC2 on colony forming ability at 200 and 300 �g/ml. There is actually a highly significant inhibitory effect at both concentrations. Presumably, the pharmacological effect is already maximized at 200 �g/ml and no further inhibition is seen at a slightly higher concentration, so this is not surprising. As stated in the text PC2 has a lesser effect than that observed with psiadin. On the other hand, Fig 4a and b show the effects of psiadin and PC2 on NF-�B activity. The reviewer is correct that PC2 (in contrast to psiadin) has no effect here. But two completely different things are being measured in these two figures so we do not see any problem or inconsistency.

9. Result headings should describe the major results discussed. For example, measuring ROS and protein expression are not appropriate.

Response: Results headings were modified to describe the major results discussed. 

10. There should be a clear demarcation of results and discussion to increase the readability of the work.

Response: The journal guidelines allow results and discussion sections to be combined, which is our preferred way of presentation. The authors hope that this is fine.

Reviewer 3

1. Overall, this is a concise, clear, well planned and well executed study. The manuscript is well-written. The study focused on anticancer potential of three scarce terpenes, psiadin, plectranthone and saudinolide. Their anticancer activity was tested on two human colorectal carcinoma cell lines, CCL233 and CCL235.

The presentation of data is clear and concise in text and figures. Results have been explained comprehensively and discussed critically to reach conclusions. Bibliography contains complete, accurate and up-to-date list of references with cautious care given to research ethics, including plagiarism and proper citation. This is a good quality manuscript but some MAJOR REVISIONS are required before it is accepted for publication.

Response: It was not clear what the reviewer meant by a “Major revisions”. However, after inquiring about that, the editor contacted the reviewer who explained that he meant synergism study (point 2 below).

2. Please also include synergism (if any) between the terpenes tested in this study with standard drug: the authors should check the combined effect of psiadin and plectranthone, psiadin and saudinolide, plectranthone and saudinolide, in comparison with some clinically available anticancer drugs.

Response: Synergism between the tested terpenes and clinically available drugs has not been tested in the current study. Although such a study would have enriched the findings of the current project, it is beyond the proposed aims of this project. 

The synergism will be considered in future study.

References used in the above responses:

1. Pelicano H, Carney D, Huang P. ROS stress in cancer cells and therapeutics implications. Drug Resist Updat. 2004;7:97–110. doi: 10.1016/j.drup.2004.01.004.

2. Hu W, Lee SK, Jung MJ, Heo S, Hur JH, Wang MH. Induction of cell cycle arrest and apoptosis the ethyl acetate fraction of Kalopanax pictus leaves in human colon cells. Bioresour Technol. 2010;101:9366–72. doi: 10.1016/j.biortech.2010.06.091.

3. Jaganathan SK. Growth inhibition by caffeic acid, one of the phenolic constituents of honey, in HCT15 colon cancer cells. Sci World J. 2012;2012:372345. doi: 10.1100/2012/372345. 

4. Khan M, Li T, Ahmed Khan MK, Rasul A, Nawaz F, Sun M, et al. Alantolactone induces apoptosis in HepG2 cells through GSH depletion, inhibition of STAT3 activation, and mitochondrial dysfunction. Biomed Res Int. 2013;2013:719858.

5. Lee RT, Collins T. Nuclear factor-kB and cell survival: IAPs call for support. Circ Res. 2001;88:262–4. doi: 10.1161/01.RES.88.3.262.

6. Levkau B, Garton KJ, Ferri N, Kloke K, Nofer J-R, Baba HA, Raines EW, Breithardt G. × − IAP induce cell-cycle arrest and activates nuclear factor-kB: new survival pathways disabled by caspase-mediated cleavage during apoptosis of human endothelial cells. Circ Res. 2001;88:282–90. doi: 10.1161/01.RES.88.3.282.

---

## [Editor Report · Decision Letter 1]

24 May 2021

Psiadin and Plectranthone selectively inhibit colorectal carcinoma cells proliferation via modulating cyclins signaling and apoptotic pathways

PONE-D-21-06574R1

Dear Dr. Khaled Y Orabi,

We’re pleased to inform you that your manuscript has been judged scientifically suitable for publication and will be formally accepted for publication once it meets all outstanding technical requirements.

Kind regards,

Mahmood ur Rahman Ansari, Ph.D

Academic Editor

PLOS ONE
---

## [Editor Report · Acceptance letter]

26 May 2021

PONE-D-21-06574R1 

Psiadin and Plectranthone selectively inhibit colorectal carcinoma cells proliferation *via* modulating cyclins signaling and apoptotic pathways 

Dear Dr. Orabi:

I'm pleased to inform you that your manuscript has been deemed suitable for publication in PLOS ONE. Congratulations! Your manuscript is now with our production department. 

Kind regards, 

on behalf of

Dr. Mahmood ur Rahman Ansari 

Academic Editor

PLOS ONE